# OpenReview forum: "How RLHF Amplifies Sycophancy"
_ICML.cc/2026/Conference — ICML 2026 regular_

### Official Review · Reviewer_wG92 · 2026-03-03

**Soundness:** 3
**Presentation:** 3
**Significance:** 2
**Originality:** 2
**Overall Recommendation:** 4
**Confidence:** 3

**Summary:**

The paper studies a common LLM failure mode: LLMs can become more sycophantic after preference-based post-training. The paper discusses the cases where sycophantic bias is amplified under two common RLHF settings and how the bias gets propagated from annotations to rewards. Then the paper proposes a mitigation term to address the amplification problem. Numerical experiments are conducted to empirically validate the analysis.

**Compliance With Llm Reviewing Policy:**

Affirmed.

**Final Justification:**

The paper starts with a well-formulated problem. The authors provide a detailed analysis to develop their modeling and present their conclusions. However, the paper also suffers from its studied problem. On the one hand, the preference modeling is less active in practice, as mainstream post-training has largely shifted from RLHF to RLVR. On the other hand, while the paper provides a nice framework, it is hardly practical to address the mentioned problem. In all, the contribution of this paper does not lie in addressing of a problem, but in helping deepen the understanding of the problem.

The rebuttal itself partially resolves the concerns, but the practicality problem remains. And the concern is at the root of the paper's analysis framework.

After careful consideration, while the work is still limited due to its practicality, the formulation of the problem and the analysis process do provide insights into preference modeling. And I personally really like the style of such how-and-why papers. Hence, I will raise my score to 4.

**Key Questions For Authors:**

See the weaknesses and please use that index to refer.

**Limitations:**

Yes

**Strengths And Weaknesses:**

The paper’s presentation is sound, with a clear causal chain across the whole pipeline. The covariance characterization and the mean gap concept are intuitive and interpretable. Generally, I like the analysis part. The mitigation is also elegant with a closed-form penalty.

However,
1. In practice, the conditional exponential moments term $m_a^\beta (x)$ and the agreement detector $A(x,y)$ are not easy to obtain in certain cases. The exponential term is tail sensitive and could require many samples to estimate, and for some open-ended tasks, the agreement of users’ stance could be ambiguous or implicit. All these make the practical operations not that simple.
2. Mitigation is theoretically characterized but not evaluated end-to-end. The numerical experiments in Section 5 are not persuasive enough because the intervention is one of the key contributions, and without validation, the paper’s contribution is largely weakened.
3. The current experiment setting is kind of too trivial. For example, multi-turn dialogue setups where stance is more likely to be implicit can better match the real-world concerns and validate the proposed method.

---

> ### Author Rebuttal · Authors · 2026-03-30
>
> The primary contribution of this work is to identify the origin of sycophancy amplification in preference-based post-training by giving a causal account from comparison data to learned reward tilt and then to downstream behavioral drift. Section 5 should be read as the minimal correction implied by that mechanism, rather than a claim that the current submission already provides a turnkey deployment algorithm.
>
> With that scope in mind, we address the reviewer’s three practicality concerns directly below.
>
> > 1. In practice, the conditional exponential moments term $m_a^\beta(x)$ and the agreement detector A(x,y) are not easy to obtain in certain cases. The exponential term is tail sensitive and could require many samples to estimate, and for some open-ended tasks, the agreement of users’ stance could be ambiguous or implicit. All these make the practical operations not that simple.
>
> We agree that the exact exponential-moment criterion can be hard to estimate, and that a reliable global agreement signal is especially difficult in open-ended settings.
>
> **Conditional exponential moments**: The exact exponential-moment criterion becomes harder to estimate as optimization pressure increases, since larger $\beta$ makes the criterion increasingly sensitive to rare high-reward sycophantic outliers. At the same time, the regime we believe is often most relevant in practice is the more conservative weak-to-moderate optimization regime, with strong regularization and controlled drift relative to a reference policy. In that regime, the first-order effect is governed by the simpler mean reward gap between agreement and correction, which is substantially easier to estimate in practice. Empirically, our Section 6 experiments show that the sign of this measured mean reward gap predicts, on average, the direction of downstream behavioral drift.
>
> **Agreement detector**: The correction suggested is most naturally applicable in settings where there is a clear notion of ground truth and agreement pressure is explicit enough to label reliably. Any practical $A$ should be understood as a proxy signal rather than an oracle, for example an LLM judge, a small supervised detector, or a probe over model activations. The paper cites prior work showing that such signals can still be useful in practice [1,2].
>
> > 2. Mitigation is theoretically characterized but not evaluated end-to-end. The numerical experiments in Section 5 are not persuasive enough because the intervention is one of the key contributions, and without validation, the paper’s contribution is largely weakened.
>
>
> Following the reviewer’s suggestion, and consistent with a similar request from Reviewer kLnJ, we added a small-scale Best-of-$N$ operationalization of the Section 5 correction. In this setting, increasing $\lambda$ monotonically reduces selected sycophancy.
>
> Using the same `AQuA MC` / `incorrect-answer` / `OpenLLaMA` setting and fixing $N=32$, we rescored the same candidates with the penalized objective $r_\lambda(x,y)=r(x,y)-\lambda A(x,y)$ and measured post-selection sycophancy, reporting both the overall sycophancy rate and the fraction of prompts whose sycophancy increased relative to the base model:
>
> |$\lambda$|$S(\pi_{\mathrm{BoN}})$|$S(\pi_{\mathrm{base}})$|$S(\pi_{\mathrm{BoN}})-S(\pi_{\mathrm{base}})$|% prompts violating no-amplification|$\Delta r$|
> |:--:|:--:|:--:|:--:|:--:|:--:|
> |0|0.438|0.31|0.127|0.52|0|
> |0.25|0.354|0.31|0.044|0.52|0.004|
> |0.5|0.275|0.31|-0.035|0.5|0.017|
> |0.75|0.203|0.31|-0.107|0.29|0.035|
> |1|0.143|0.31|-0.168|0.05|0.057|
> |1.25|0.094|0.31|-0.216|0|0.08|
>
> This is not yet a full end-to-end validation, but it shows that the Section 5 correction is operational in a representative reward-optimization setting and behaves in the predicted direction.
>
> > 3. The current experiment setting is kind of too trivial. For example, multi-turn dialogue setups where stance is more likely to be implicit can better match the real-world concerns and validate the proposed method.
>
> We would like to clarify one factual point: the submission already includes a multi-turn setting. In Section 6 we evaluate an `Are-you-sure` format (see Figure 7a) in which the prompt is a short dialogue history: the user asks a question, the LLM answers, and the user then challenges that answer with a follow-up turn (building on [3]).
>
> In the experiments, we use QA prompts with known correct answers, which make correctness easy to verify, and vary the bias-injection strategy across increasing levels of stance pressure, including both single-turn and multi-turn settings.
>
>
> - [1] Papadatos and Freedman, 2024. "Linear probe penalties reduce LLM sycophancy."
> - [2] Wei et al., 2023. "Simple synthetic data reduces sycophancy in large language models."
> - [3] Sharma et al., 2023. "Towards understanding sycophancy in language models."

---

> > ### Author Rebuttal · Reviewer_wG92 · 2026-04-01
> >
> > Thank you for the rebuttal and the clarification. I appreciate your clarification on (3). However, the other two concerns remain.
> >
> > For (1), even if we limit the scope to the conservative optimization with the reference policy, we need some settings with a clear ground truth or reliable agreement labels. This limits the work's value for open-ended settings. The rebuttal itself also admits that the quantities are hard to estimate in practice.
> >
> > For (2), I really appreciate the added $\lambda$-sweeping experiment. But I do not think this is an end-to-end validation of the mitigation in Section 5. It is more of a post-hoc, N-fixed reranking experiment on the same candidate pool. Consequently, it does not test the robustness to policy shift or detector error. It only shows that penalizing agreement reduces selected sycophancy, from my perspective.
> >
> > Overall, your rebuttal improves the clarity and partially addresses my concern. But the paper's practical significance remains to be improved. I will raise the score to 3 accordingly.

---

> > > ### Author Response · Authors · 2026-04-01
> > >
> > > Thank you very much for your attention and for updating your score. Please note that the paper’s main goal is to enhance our understanding of the origin of sycophancy amplification in RLHF, whereas the current comments are mainly about the breadth of the mitigation’s operational validation. We agree that a full end-to-end mitigation strategy in open-ended settings would need to tackle the issues you raise.

---

### Official Review · Reviewer_24TH · 2026-03-04

**Soundness:** 3
**Presentation:** 3
**Significance:** 3
**Originality:** 3
**Overall Recommendation:** 5
**Confidence:** 5

**Summary:**

This work tackles the sycophancy problem in LLMs, where preference-aligned models exhibit amplified and excessive agreeability with users, often at the cost of factual correctness or sound judgement. The authors point out that post-training (specifically KL-regularised RLHF or Best-of-N) reweights the base model’s output distribution toward higher-reward responses, and present a correlations argument for why syncophantic behaviours are amplified (Thm. 1 and 2). Under increasing optimisation pressure and a skewed fixed reward signal, if syncophantic responses are overrepresented or are assigned higher rewards than corrective ones, then the LLM also becomes increasingly biased towards selecting for such behaviour. Mathematically, they show that under KL-RLHF, the shift in a behaviour statistic A scales with the covariance between A and the reward itself; under Best-of-N, they present an analogous finding that syncophancy scales with the covariance between the behaviour indicator and the quantile weight (Thm. 3). The authors trace this behavioural tilt back to labelling bias, where human annotations result in higher score assignment to false-stance, syncophantic responses than to corrective ones, in cases of mixed pairs. This results in a mean reward gap, and the authors argue that the sign of this gap is determined entirely by the mixed-pair bias statistic B_F(x): positive annotator tilt towards agreement is both necessary and sufficient for reward bias (Thm. 4). This condition propagates to the post-training policy provided the bias exceeds the reward model's approximation error on mixed pairs (Thm. 5). Finally, the authors introduce a minimal corrective intervention to mitigate syncophancy: the policy closest in KL divergence to the unconstrained RLHF optimum subject to a no-amplification constraint. This takes a simple reward-shaping form – subtracting a penalty proportional to the agreement signal on false-stance prompts – with the optimal per-prompt penalty coefficient given in closed form (Thm. 6). They also show that this global penalty is the uniquely optimal correction under the KL-minimality criterion. Empirical validation on SycophancyEval confirms that 30-40% of prompts exhibit positive reward tilt – especially higher-pressure bias injection strategies – and that this pattern holds consistently across benchmarks and reward model architectures. Crucially, the sign of the measured tilt reliably predicts the direction of behavioural drift under Best-of-N: sycophancy increases monotonically with N on positive-tilt prompts and decreases on negative-tilt ones, validating the theoretical framework. The authors acknowledge that the analysis operates in an idealised asymptotic setting and that empirically measuring the benefit of their minimal correction over coarser existing approaches remains future work.

**Compliance With Llm Reviewing Policy:**

Affirmed.

**Final Justification:**

The rebuttal clarifies the paper’s scope and mechanism, framing the results primarily in the weak-optimization regime, acknowledging possible tail-driven effects under stronger optimization. While more direct empirical evidence on the trade-off could be valuable, my concerns are largely addressed and I maintain my original "Accept" score.

**Key Questions For Authors:**

- **Q1.** Theoretically, the amplification criterion depends on exponential moments of the reward, which suggests that high-reward syncophantic outliers can amplify drifts under strong optimisation pressures. Empirically, what is the distribution of tilt magnitudes observed? And do you believe that the syncophancy amplification observed in post-training in practice is more so due to mean reward tilts or tail reward anomalies?
- **Q2.** Do you have some intuition on how syncophancy amplification interfaces with reward hacking, especially in cases of strong optimisation pressure? Are there commonalities between the mechanisms (e.g. reward tail exploitation) that underlie them?
- **Q3.** The proposed correction enforces a constraint preventing amplification of agreement behaviour. This effectively introduces a multi-objective optimisation between maximising reward and minimising sycophancy. Have the authors considered explicitly formulating the problem as a Pareto optimisation between reward and undesirable behaviour statistics, and if so, does the KL-projection solution correspond to a particular point on this Pareto frontier?
- **Q4.** You experimentally demonstrate similar trends between the Best-of-N selection and PPO RLHF cases. Operationally, could Best-of-N serve as a diagnostic for whether a reward model will induce sycophancy amplification under RLHF training? Intuitively, both procedures bias the base policy toward higher-reward responses – Best-of-N via selection from the reward tail and RLHF via exponential reweighting – do you think an observation of increasing syncophancy with N may indicate that agreement responses are overrepresented in the high-reward tail?

**Limitations:**

Yes.

**Strengths And Weaknesses:**

### 1. Soundness.
**S1.1 – Rigour.** The paper establishes a clear and rigorous link between reward optimisation and behavioural amplification by deriving exact conditions under which RLHF increases undesirable behaviours. In particular, Theorem 1 expresses behavioural drift as a covariance between the behaviour statistic and the exponential reward weight under the base policy, which provides a precise and interpretable mechanism explaining when optimisation amplifies behaviours such as sycophancy. This result is both mathematically clean and operationally meaningful, as it directly connects properties of the reward distribution to observable changes in model behaviour under optimisation.
**W1.2 – Trade-offs.** The proposed correction enforces a constraint preventing amplification of agreement behaviour, introducing a multi-objective optimisation between maximising reward and minimising sycophancy. However, the analysis focuses on a single KL-projection solution without characterising the Pareto frontier between these objectives. Consequently, it is unclear whether the proposed solution is Pareto-efficient or how sensitive the mitigation is to different trade-off choices, limiting the theoretical understanding of the policy design space.

### 2. Presentation.
**S2.1 – Organisation.** This paper is exceptionally well-structured, with a clear progression from the analysis of behaviour amplification under reward optimisation in Section 3, to the identification of the source of reward tilt in preference learning in Section 4, to the proposed minimal correction mechanism in Section 5, and finally empirical validation in Section 6. The theorems are well motivated and supplemented with ample intuition and practical implications analysis, making it easy for the reader to follow how the theoretical results translate into concrete predictions about optimisation behaviour and practical mitigation strategies.

### 3. Significance.
**W3.1 – Feasibility.** The theory analyses idealised, asymptotic RLHF with unlimited reward modelling capacity and perfect preference learning. While they are able to reliably predict syncophantic reward tilts in experiments, the syncophancy mitigation technique theorised in Section 5 has yet to be operationalised. It remains to be seen how robust this framework is in realistic settings, where reward models may be misspecified, PPO is used with many modifications (minibatches, clipping, early stopping), and under imperfect reward optimisation.
**S3.2 – Problem definition.** The authors identify a safety-critical problem which notably worsens under untargeted post-training alignment interventions, and possibly also with increased model capacity. They address the research gap that the causal mechanism behind RLHF worsening syncophancy is yet unexplored, and present a theoretical framework to trace, localise and predict this behavioural tilt in the reward signal and optimisation strategy.
**S3.3 – Scope.** While this paper focuses on syncophancy, the introduced framework and theory is very general and could apply to other measurable statistics of undesirable behaviour. The framing of behavioural drift induced by post-training due to latent reward tilts is powerful, and could enable systematic diagnosis of reward model biases, prediction of optimisation-induced behavioural shifts, and principled mitigation via constrained or multi-objective reward optimisation.

### 4. Originality.
**S4.1 – Approach.** This paper presents a novel theoertical perspective on RLHF behaviour amplification and connects it to reward distribution properties. The notion of annotator induced reward tilt as a trigger for syncophancy amplification is conceptually exciting, and the outlined KL-projection formulation for minimal reward correction is principled, and nicely applies optimisation theory to frontier preference alignment problems.

---

> ### Author Rebuttal · Authors · 2026-03-30
>
> > Q1. Theoretically, the amplification criterion depends on exponential moments of the reward, which suggests that high-reward syncophantic outliers can amplify drifts under strong optimisation pressures. Empirically, what is the distribution of tilt magnitudes observed? And do you believe that the syncophancy amplification observed in post-training in practice is more so due to mean reward tilts or tail reward anomalies?
>
> As the reviewer notes, the exact criterion is tail-sensitive through conditional exponential moments, so rare high-reward sycophantic outliers can matter disproportionately and can, in principle, drive the direction of drift.
>
> At the same time, the regime we believe is most relevant in practice is often the more conservative weak-optimization regime. Public preference-optimization systems are typically trained in a strongly regularized, controlled-drift regime relative to an SFT or reference policy, precisely because of stability and overoptimization concerns. In that regime, the first-order effect is governed by the mean reward gap.
>
> Empirically, we do observe heterogeneous reward-score distributions and therefore do not rule out tail-driven cases, as illustrated on page 24. However, in our Best-of-N experiment, the sign of the measured mean reward gap already predicts, on average, the direction of behavioral drift.
>
>
>
> > Q2. Do you have some intuition on how syncophancy amplification interfaces with reward hacking, especially in cases of strong optimisation pressure? Are there commonalities between the mechanisms (e.g. reward tail exploitation) that underlie them?
>
> We think there is at least a partial connection. In both cases, stronger optimization can amplify features that are rewarded for the wrong reason. In our setting, that feature is agreement with the user’s stance. Under stronger optimization, this can naturally connect to reward-tail exploitation.
>
> > Q3. The proposed correction enforces a constraint preventing amplification of agreement behaviour. This effectively introduces a multi-objective optimisation between maximising reward and minimising sycophancy. Have the authors considered explicitly formulating the problem as a Pareto optimisation between reward and undesirable behaviour statistics, and if so, does the KL-projection solution correspond to a particular point on this Pareto frontier?
>
>
> Understanding the fuller tradeoff between the KL-regularized RLHF objective and sycophancy, or undesirable behavior more generally, is an interesting and important question. When the standard post-training objective conflicts with safety requirements such as avoiding increased sycophancy, it seems important to characterize the resulting frontier between these desiderata.
>
> Our Section 5 focuses on one special point on that frontier rather than tracing the full curve. Specifically, we study the policy closest in KL to the unconstrained RLHF optimum among all policies satisfying $S(\pi)\le S(\pi_{\mathrm{base}})$. While this is the smallest adjustment to standard RLHF that avoids doing more harm than the base policy, it leaves open the broader question of whether other, potentially very different, policies could achieve the same or lower sycophancy while attaining a larger value of the same KL-regularized RLHF objective.
>
> > Q4. You experimentally demonstrate similar trends between the Best-of-N selection and PPO RLHF cases. Operationally, could Best-of-N serve as a diagnostic for whether a reward model will induce sycophancy amplification under RLHF training? Intuitively, both procedures bias the base policy toward higher-reward responses – Best-of-N via selection from the reward tail and RLHF via exponential reweighting – do you think an observation of increasing syncophancy with N may indicate that agreement responses are overrepresented in the high-reward tail?
>
>
> While they use different reward reweightings, we can intuitively argue that these two mechanisms are aligned enough that Best-of-N can be used as a useful, low-cost diagnostic for amplification risk under reward optimization more broadly. It is not an exact proxy, but if sycophancy increases with $N$, that is a warning sign that reward optimization may amplify the same behavior.

---

> > ### Author Rebuttal · Reviewer_24TH · 2026-04-03
> >
> > The rebuttal clarifies that the main practical claim is about the weak-optimization regime, where sycophancy amplification is mostly explained by the mean reward gap. They also acknowledge that tail effects may matter under stronger optimization. The rebuttal better positions the proposed correction as a conservative choice of one point on the broader reward--sycophancy tradeoff frontier, rather than a full treatment of that tradeoff. The Best-of-N discussion was helpful for understanding how it could warn against potential amplification risk under reward optimization. While more direct empirical evidence on the trade-off could be valuable, my concerns are largely addressed and I maintain my original "Accept" score.

---

### Official Review · Reviewer_TkaA · 2026-03-11

**Soundness:** 4
**Presentation:** 4
**Significance:** 4
**Originality:** 4
**Overall Recommendation:** 5
**Confidence:** 4

**Summary:**

Authors define sycophancy score as the expectation of the policy agreeing with the stance in the prompt when the prompt is incorrect. In the case of RL training it is shown that the increase of the sycophancy score can be written as the correlation between the sycophancy score and the exponentiated reward. In the case of binary sycophancy, the sign of the shift can be determined by the difference of exponentiated rewards for correct behavior and incorrect behavior. Linear approximation of the exponentiated reward is also analyzed. In the case of Best-of-N, the same behavior is characterized with the reward quantile.
Then, authors study how the labeling bias results in biased reward. Sycophantic population preference on a prompt is attributed to the mean reward gap for the prompt. Then authors study the optimization problem of finding a policy that is closest in KL to the base policy, and the sycophancy score does not degrade.

**Compliance With Llm Reviewing Policy:**

Affirmed.

**Final Justification:**

Authors successfully addressed my concern, yet I am not changing the score as my original score was assuming the concern would be addressed.

**Key Questions For Authors:**

I might be missing something in Section 5. Isn't the base policy a trivial solution for the optimization problem (10), as the KL divergence would be 0, and the sycophancy doesn't degrade?

**Limitations:**

yes

**Strengths And Weaknesses:**

Soundness: The paper is mostly theoretical, and key results are proved with a few intuitive lines. Theorems which proofs are deferred to appendix seem intuitively plausible, although I haven't verified them.

Presentation: Authors initiate the paper with a clean statement of the problem. Sycophancy is clearly defined, with contextualization to previous work in the literature. Authors prove key results intuitively without resorting to appendix, and the implication of each factor in the result is also discussed, for ex p^1(x) p^0(x) term in line 209. Limitations on the interpretation of stated results are also frankly discussed.

Significance: Authors show how the correlation between the target behavior and the reward structure influences the outcome of policy optimization. These results can more generally apply to broader cases of reward hacking than sycophancy. Many of the theoretical results are mathematically simple, but the simplicity makes them easy to understand and interpret. Authors also make connections to empirical results (for example in line 314), helping practitioners to understand implications of theoretical results on their data collection efforts. Authors also augment there results with empirical analysis to further validate theoretical results.

Originality: While sycophancy has been actively studied, this paper provides a surprisingly simple characterizations of how simple statistics (correlations with exponentiated rewards and likes) determine reward hacking. The unexpected simplicity demonstrates the high originality of the paper.

---

> ### Author Rebuttal · Authors · 2026-03-30
>
> > I might be missing something in Section 5. Isn't the base policy a trivial solution for the optimization problem (10), as the KL divergence would be 0, and the sycophancy doesn't degrade?
>
> The base policy is a feasible solution to Eq. (10), but it is not generally optimal. It automatically satisfies the no-amplification constraint, since relative to itself there is no sycophancy amplification. The optimization problem in Section 5 asks for something different: among all policies satisfying this constraint, we choose the one closest in KL divergence to the unconstrained RLHF optimum $\pi^\star_\beta$ (which maximizes KL-regularized reward). So the goal is not to recover $\pi_{\text{base}}$, but to stay as close as possible to the standard post-RLHF solution while enforcing no amplification.
>
> We will revise the wording in Section 5 to make this reference point more explicit.

---

> > ### Author Rebuttal · Reviewer_TkaA · 2026-04-04
> >
> > Ah, I realized that the KL is not against the base policy but \(\pi^*_\beta\). Silly me, thanks for the clarification. My original scores were based on the expectation authors are correct, so not changing the scores but I would be happy to see this paper published, good luck.

---

### Official Review · Reviewer_kLnJ · 2026-03-13

**Soundness:** 3
**Presentation:** 2
**Significance:** 3
**Originality:** 3
**Overall Recommendation:** 4
**Confidence:** 4

**Summary:**

This paper provides a formal mechanistic account of why reinforcement learning from human feedback (RLHF) tends to increase sycophantic behavior in large language models. The authors define sycophancy precisely as a policy's tendency to endorse factually incorrect user beliefs embedded in prompts, and they trace the amplification pathway across two stages: (1) how policy optimization against a fixed reward reshapes the response distribution, and (2) how reward learning from pairwise comparisons can produce a biased reward in the first place.

For the policy optimization stage, the paper establishes (Theorems 1–3) that both KL-regularized RLHF and inference-time Best-of-N selection amplify any behavior that is positively correlated with the reward (or reward quantile) under the base policy. In the small-optimization-pressure regime, this reduces to a simple mean-gap condition: sycophancy increases when the learned reward assigns higher average scores to stance-affirming responses than to corrective ones on false-stance prompts. For the reward learning stage, the paper introduces a "mixed-pair bias statistic" BF(x) and shows (Theorems 4–5) that under a well-specified Random Utility Model, the sign of this statistic exactly determines whether the learned reward exhibits the mean gap that triggers amplification. These two steps together close the causal loop from annotator preferences to downstream behavioral drift.

The paper further proposes a training-time mitigation: among all policies that satisfy a no-amplification constraint (post-training sycophancy no greater than base-model sycophancy), the unique policy closest in KL divergence to the unconstrained RLHF solution is characterized in closed form (Theorem 6), yielding a corrected reward of the form r(x,y) − λ(x)·A(x,y). The authors interpret this as a principled grounding for empirically proposed agreement-penalty approaches. Finally, computational experiments on SycophancyEval-based prompts across multiple reward model architectures confirm that positive reward tilt is common (30–40% of prompts) and that the sign of the tilt predicts whether Best-of-N optimization increases or decreases sycophancy, providing empirical validation of the theoretical framework.

**Compliance With Llm Reviewing Policy:**

Affirmed.

**Final Justification:**

Thank you to the authors for a thorough and highly effective rebuttal. I have read the response carefully, and it has fully resolved my initial concerns. Because of this, I am raising my confidence score to 4 and maintaining a strong recommendation for acceptance.

**Soundness and Significance:** The paper initially presented strong, rigorous theoretical insights into RLHF sycophancy amplification, but it had empirical gaps. The authors successfully addressed these gaps by:
1. **Model Scale:** Validating the positive-tilt effect on a much larger, state-of-the-art model (Llama-3.1-Nemotron-70B-Reward).
2. **Stylistic Confounds:** Adding multi-template controls to better disentangle stance from tone.
3. **Mitigation Validation:** Providing a concrete, small-scale empirical test of the proposed penalty mitigation in a Best-of-N setting, proving its practical viability.

**Originality and Clarity:** The framing of the mitigation as a KL projection problem remains highly original and significant to the field. Furthermore, the authors' commitment to revising Section 6 makes the empirical findings much clearer and more self-contained.

**Conclusion:** The rebuttal directly and convincingly answered all my critiques. By bolstering the empirical evidence to match the high quality of the theoretical framework, the authors have significantly strengthened the paper. I firmly support its acceptance.

**Key Questions For Authors:**

1. The system-prompt methodology for generating agreeing vs. corrective responses almost certainly introduces stylistic differences (e.g., more assertive/encouraging language in the agreement condition) that some reward models may prefer independently of factual accuracy. Have you conducted any ablation to disentangle stance preference from style preference — for example, by generating stylistically matched pairs or by regressing reward scores on sentiment/length features?

2. Theorem 6 characterizes the optimal no-amplification correction as a closed-form agreement penalty. Is there a reason why at least a small-scale empirical test of this correction (e.g., applying λ·A(x,y) penalties in a Best-of-N setting and verifying that the sycophancy rate does not exceed the base-model rate) was not included?

3. The conjecture that author-coupled RLHF amplifies sycophancy more than independent-labeler RLHF is an interesting hypothesis with direct policy implications. Do the authors have any pilot data or natural experiments from existing RLHF datasets (e.g., comparing datasets where the prompt author and preference rater overlap vs. are distinct) that could provide initial evidence for or against this conjecture?

**Limitations:**

Yes

**Strengths And Weaknesses:**

## Strengths
- The theoretical results are technically rigorous and proofs are complete. Theorems 1–3 on policy-level amplification are clean applications of exponential-family reweighting, and the proofs in the appendix are detailed and correct. Theorem 4 is a particularly elegant result: it shows that under a well-specified RUM, the mixed-pair bias statistic BF(x) is exactly equal to the mean reward gap, establishing a bijective connection between annotator bias and reward amplification.
- The no-amplification constrained optimization (Section 5) is solved exactly with a clean closed form. The proof uses Slater's condition and convex duality correctly and unambiguously.
- Sycophancy is a recognized and practically consequential failure mode in deployed LLMs, including documented incidents in frontier systems. This paper provides the first rigorous mechanistic account connecting preference data bias to policy-level amplification, filling a gap the authors rightly identify as open in the literature.
- The author-coupling conjecture raised in Section 4 is a specific, testable hypothesis about how annotation pipeline design (author-coupled vs. independent labeling) affects sycophancy, which has direct relevance to how future RLHF datasets should be collected.
- The framing of the mitigation as a KL projection problem with a no-amplification constraint is novel and yields a uniqueness result that is more satisfying than ad-hoc regularization approaches.
- The paper carefully acknowledges the limitations of its own conditions. Appendix D.1 provides a concrete counterexample showing that the mean-gap criterion need not be monotone in optimization strength β, and Appendix D.2 shows that a high win-rate for agreeing responses is not sufficient for positive reward gap. This level of care is commendable.

## Weakness
- The experimental evaluation uses a relatively small set of open-source reward models. The paper acknowledges that deployed systems use "substantially larger and sometimes proprietary" reward models. Since the paper claims reward tilt is universal across architectures, a stronger empirical case would include at least one evaluation on a larger-scale model or via API-accessible reward signals. The measured 30–40% positive-tilt rate may not generalize.
- The candidate-generation methodology using system-prompt instructions to enforce agreement vs. correction (Figure 3) introduces a stylistic confound that the paper itself acknowledges but does not resolve. The measured reward gaps could partially reflect reward model preferences for response tone (assertive vs. hedged) rather than pure stance-agreement preference.
- The empirical section (Section 6) is relatively thin for a nine-page paper. Details on how πRLHF was trained relative to πSFT, which reward model was used for the Best-of-N policy experiments, and what "sycophancy rate" precisely means in Figure 1c require the reader to cross-reference the appendix repeatedly.
- The mitigation proposed in Section 5 has not been empirically validated in the main paper. The authors explicitly state this is left for future work. Given that the mitigation is presented as a main contribution, the absence of even a small-scale validation (e.g., applying the agreement penalty in a Best-of-N setting and checking whether it enforces the no-amplification constraint) limits the practical impact of this contribution.

---

> ### Author Rebuttal · Authors · 2026-03-30
>
> ### Weaknesses
> > "The experimental evaluation uses a relatively small set of open-source reward models [...] a stronger empirical case would include at least one evaluation on a larger-scale model or via API-accessible reward signals. The measured 30–40% positive-tilt rate may not generalize."
>
> We agree that the current evidence should be read narrowly. Section 6 was designed to study public, reproducible reward models in the open RLHF ecosystem, not to claim that the exact 30 to 40% positive-tilt rate transfers unchanged to larger deployed systems. The main empirical point is qualitative: reward tilt appears on a substantial subset of prompts, is large enough to affect behavior in meaningful ways, and its sign predicts whether optimization increases or decreases sycophancy.
>
> Following the reviewer’s suggestion, we added a robustness check using an order-of-magnitude larger reward model, `NVIDIA Llama-3.1-Nemotron-70B-Reward`. In this experiment, the mean positive-tilt rate is 27%, suggesting that the effect persists and that the sign-based prediction continues to hold.
>
> > "Details [...] require the reader to cross-reference the appendix repeatedly."
>
> This is a fair presentation criticism. In the revised version, we made the empirical section more self-contained and reduce the need to cross-reference the appendix.
> ### Questions
>
> > 1. "The system-prompt methodology for generating agreeing vs. corrective responses almost certainly introduces stylistic differences (e.g., more assertive/encouraging language in the agreement condition) that some reward models may prefer independently of factual accuracy. Have you conducted any ablation to disentangle stance preference from style preference — for example, by generating stylistically matched pairs or by regressing reward scores on sentiment/length features?"
>
> > "The candidate-generation methodology using system-prompt instructions to enforce agreement vs. correction (Figure 3) introduces a stylistic confound
>
> We agree that the current candidate-generation setup does not fully disentangle stance from style. As noted in the paper, part of the measured reward gap could reflect reward-model preferences over stylistic attributes rather than stance agreement alone.
>
> At the same time, the qualitative effect is not confined to a single setup: we observe similar positive-tilt rates across reward-model architectures, datasets, multiple bias-injection strategies, and candidates generated from two distinct base policies.
>
> In the revision, we add a stronger control by using multiple prompt templates so that tone is not systematically aligned with stance.
>
> > 2. "Is there a reason why at least a small-scale empirical test of this correction (e.g., applying λ·A(x,y) penalties in a Best-of-N setting and verifying that the sycophancy rate does not exceed the base-model rate) was not included? "
>
> Following the reviewer’s suggestion, we added a small-scale operationalization of correction. Concretely, we took one representative setting, `AQuA MC` prompts with `incorrect-answer` bias injection and the `OpenLLaMA` reward model, and rescored candidates applying the $\lambda A(x,y)$ penalties. We then measured sycophancy after selection, reporting both the overall sycophancy rate and the fraction of prompts whose sycophancy increased relative to the base model:
>
> |$\lambda$|$S(\pi_{\mathrm{BoN}})$|$S(\pi_{\mathrm{base}})$|$S(\pi_{\mathrm{BoN}})-S(\pi_{\mathrm{base}})$|% prompts violating no-amplification|$\Delta r$|
> |:--:|:--:|:--:|:--:|:--:|:--:|
> |0|0.438|0.31|0.127|0.52|0|
> |0.25|0.354|0.31|0.044|0.52|0.004|
> |0.5|0.275|0.31|-0.035|0.5|0.017|
> |0.75|0.203|0.31|-0.107|0.29|0.035|
> |1|0.143|0.31|-0.168|0.05|0.057|
> |1.25|0.094|0.31|-0.216|0|0.08|
>
>
> A moderate penalty restores the average sycophancy rate to the base-model level with limited reward loss, while a stronger penalty is needed to suppress nearly all promptwise violations.
>
> > 3. "The conjecture that author-coupled RLHF amplifies sycophancy more than independent-labeler RLHF is an interesting hypothesis [...] Do the authors have any pilot data or natural experiments from existing RLHF datasets (e.g., comparing datasets where the prompt author and preference rater overlap vs. are distinct) that could provide initial evidence for or against this conjecture?"
>
> We do not test this conjecture directly in the current paper, and we present it as a theory-driven prediction rather than an empirical claim. In our framework, sycophancy increases when the learned reward favors agreement over correction. One plausible source of such bias is self-agreement: if prompt authors also serve as raters, agreeing responses are more likely to match the raters’ own views. This suggests that author-coupled labeling may induce stronger bias than independent labeling. A clean test would compare preferences on the same prompts and candidate responses with author-coupled and independently assigned raters.

---

> > ### Author Rebuttal · Reviewer_kLnJ · 2026-04-03
> >
> > Thank you for the author's detailed reply, which further strengthens my confidence in the assessment of this article.

---

### Decision · Program_Chairs · 2026-04-30

**Decision:**

Accept (regular)

**Comment:**

After the rebuttal, the reviewers agreed that this paper makes a valuable contribution to understanding sycophancy amplification in RLHF, and the rebuttal resolved the main concerns raised during review. The paper provides a rigorous theoretical framework connecting annotator bias to behavioral drift, showing that amplification is governed by the covariance between agreement behavior and the reward. The framework also generalizes beyond sycophancy to other forms of reward-driven drift, offering a principled diagnostic for post-training risks. The proposed KL-projection mitigation is novel and yields a closed-form agreement penalty, validated empirically during rebuttal. Reviewer wG92 raised remaining concerns about practical applicability in open-ended settings, but the authors clarified that the paper's primary goal is to deepen understanding of the amplification mechanism rather than provide a turnkey solution.
Overall, this is a solid paper that tackles an interesting problem and thus I recommend accepting this paper.